# Tumor control via targeting PD-L1 with chimeric antigen receptor modified NK cells

Yvette Robbins[1], Sarah Greene[1], Jay Friedman[1], Paul E Clavijo[1],
Carter Van Waes[2], Kellsye P Fabian[3], Michelle R Padget[3], Houssein Abdul Sater[4],
John H Lee[5], Patrick Soon-Shiong[5], James Gulley[4], Jeffrey Schlom[3],
James W Hodge[3], Clint T Allen[1,6]*

[1]Translational Tumor Immunology Program, National Institute on Deafness and Other Communication Disorders, National Institutes of Health, Bethesda, United States; [2]Tumor Biology Section, National Institute on Deafness and Other Communication Disorders, National Institutes of Health, Bethesda, United States; [3]Laboratory of Tumor Immunology and Biology, National Cancer Institute, National Institutes of Health, Bethesda, United States; [4]Genitourinary Malignancies Branch, National Cancer Institute, National Institutes of Health, Bethesda, United States; [5]NantKWest, Culver City, United States; [6]Department of Otolaryngology-Head and Neck Surgery, Johns Hopkins School of Medicine, Baltimore, United States

**Abstract** Failed T cell-based immunotherapies in the presence of genomic alterations in antigen presentations pathways may be overcome by NK cell-based immunotherapy. This approach may still be limited by the presence of immunosuppressive myeloid populations. Here, we demonstrate that NK cells (haNKs) engineered to express a PD-L1 chimeric antigen receptor (CAR) haNKs killed a panel of human and murine head and neck cancer cells at low effector-to-target ratios in a PD-L1-dependent fashion. Treatment of syngeneic tumors resulted in CD8 and PD-L1-dependent tumor rejection or growth inhibition and a reduction in myeloid cells endogenously expressing high levels of PD-L1. Treatment of xenograft tumors resulted in PD-L1-dependent tumor growth inhibition. PD-L1 CAR haNKs reduced levels of macrophages and other myeloid cells endogenously expressing high PD-L1 in peripheral blood from patients with head and neck cancer. The clinical study of PD-L1 CAR haNKs is warranted.

*For correspondence:
clint.allen@nih.gov

## Introduction

T cell-based immunotherapy, such as immune checkpoint blockade or adoptive T cell transfer, is limited by the ability of T cells to detect major histocompatibility complex (MHC)-presented antigen by tumor cells. Through selective immune pressure during tumorigenesis and progression and genomic instability, subpopulations of tumor cells acquire mutations that lead to defective type II interferon (IFN) responses and altered antigen processing and presentation (*Sucker et al., 2014*; *TRACERx Consortium et al., 2017*). The presence of these mutations predicts failure to respond to immune checkpoint blockade and adoptive T cell transfer immunotherapy (*Doran et al., 2019*; *Shin et al., 2017*; *Zaretsky et al., 2016*; *Gao et al., 2016*).

Natural killer (NK) cell-based immunotherapy may overcome genetic mechanisms of resistance to T cell-based immunotherapy through antigen- and MHC-independent recognition of malignant cells. NK cells express germ-line receptors that are either stimulatory or inhibitory, and the summation of these signals determines activation status (*Caligiuri, 2008*). NK-92 cells, derived from a Non-

Hodgkin's lymphoma patient, can be continuously expanded in culture and following irradiation, can be safely administered in high doses to patients with cancer (*Gong et al., 1994*; *Williams et al., 2017*; *Tonn et al., 2013*). High-affinity NK (haNK) cells are NK-92 cells engineered to express endoplasmic reticulum-retained IL-2(11), have demonstrated potent effector function in numerous preclinical models (*Friedman et al., 2019*; *Friedman et al., 2018*), and following irradiation, can also be safely administered in high doses to patients (*Seery et al., 2019*). haNKs represent an 'off the shelf' NK cellular therapy available for pre-clinical and clinical study. However, barriers within the tumor microenvironment that further limit T cell activation and function such as the presence of immunosuppressive myeloid cells also can limit the activation and function of NK cells (*Condamine et al., 2016*; *Li et al., 2009*). Thus, addressing the presence of immunosuppressive myeloid cell populations in the periphery and tumor microenvironment of patients with cancer is likely to be required for effective NK cell-based immunotherapy.

Here, we describe the pre-clinical in vitro and in vivo study of irradiated haNK cells engineered to express a second-generation chimeric antigen receptor (CAR) targeting programmed death-ligand 1 (PD-L1). PD-L1 CAR haNKs induced cytotoxicity against human papillomavirus-positive and negative human head and neck squamous cell carcinoma (HNSCC) cells as well as murine HNSCC cells. Killing of cells by PD-L1 CAR haNKs was both direct and PD-L1 CAR-mediated as CRISPR/Cas9 knockout of PD-L1 abrogated killing within individual models by 60–90%. Treatment of immune competent mice bearing syngeneic tumors resulted in tumor rejection or growth delay that was both PD-L1 and CD8$^+$ cell dependent, demonstrating crosstalk between adaptive T cell immunity, tumor cell PD-L1 expression and sensitivity to PD-L1 CAR haNKs. Treatment also resulted in significant reduction in both peripheral and tumor infiltrating macrophages and neutrophilic and monocytic myeloid cells endogenously expressing high levels of PD-L1. Treatment of immunodeficient mice bearing human xenograft tumors also resulted in PD-L1-dependent tumor growth inhibition. The ability of PD-L1 CAR haNKs to reduce myeloid populations endogenously expressing high levels of PD-L1 was validated in co-incubation assays with peripheral blood from patients with advanced stage HNSCC. These results provide the pre-clinical data required for the phase I study of PD-L1 CAR haNKs in patients with relapsed malignancies.

## Results

haNKs were engineered to express a CAR targeting PD-L1. This second-generation CAR includes a single-chain variable fragment (scFv) derived from NANT-601, and IgG1 mAb targeting PD-L1, along with a CD8 hinge, a CD28 transmembrane domain and an intracellular FcεR1γ signaling domain (diagramed in *Figure 1—figure supplement 1A*). Expression of the PD-L1 CAR was verified on PD-L1 CAR haNK cells by flow cytometry (*Figure 1—figure supplement 1B*). The ability of irradiated PD-L1 CAR haNKs to kill a panel of HPV-positive and -negative human HNSCC cells was directly compared to that of irradiated parental haNKs that do not express the CAR (*Figure 1*). To determine if increased PD-L1 expression on target cells would increase killing capacity by PD-L1 CAR haNKs, some cells were exposed to IFNγ. Exposure to IFNγ significantly increased PD-L1 expression on all cells (left panels, *Figure 1A–E*). Exposure of cells to IFNγ also consistently increased MHC class I expression, but variably decreased expression of NKG2D ligands MHC class I polypeptide-related sequence A/B (MICA/B) on target cells (*Figure 1—figure supplement 2*). As measured by real-time impedance analysis (middle and right panels, *Figure 1A–E*), killing of HNSCC cells by PD-L1 CAR haNKs at low effector-to-target ratios (0.5:1 or 1:1) was significantly greater compared to haNKs in the absence of IFNγ pre-treatment of target cells. Pre-treatment of target cells with IFNγ significantly increased killing by PD-L1 CAR haNKs but did not increase killing by haNKs. Killing of IFNγ pre-treated human target cells by PD-L1 CAR haNKs was durable as no tumor cell growth rebound was observed. No clear correlation between HLA class I or MICA/B expression and killing by PD-L1 CAR haNKs or haNKs existed. These data suggested that at matched effector-to-target ratios, PD-L1 CAR haNKs kill multiple human HNSCC targets to a greater degree than haNKs, and that susceptibility to PD-L1 CAR haNK killing can be increased with greater PD-L1 expression following IFNγ pre-treatment of target cells.

NK cell exhaustion can limit their effector function (*Bi and Tian, 2017*). To determine the effect of prior target cell engagement on effector killing capacity in PD-L1 CAR haNK cells, these cells were assayed for their ability to lyse HPV-negative UMSCC-1 or HPV-positive UMSCC-47 targets after a

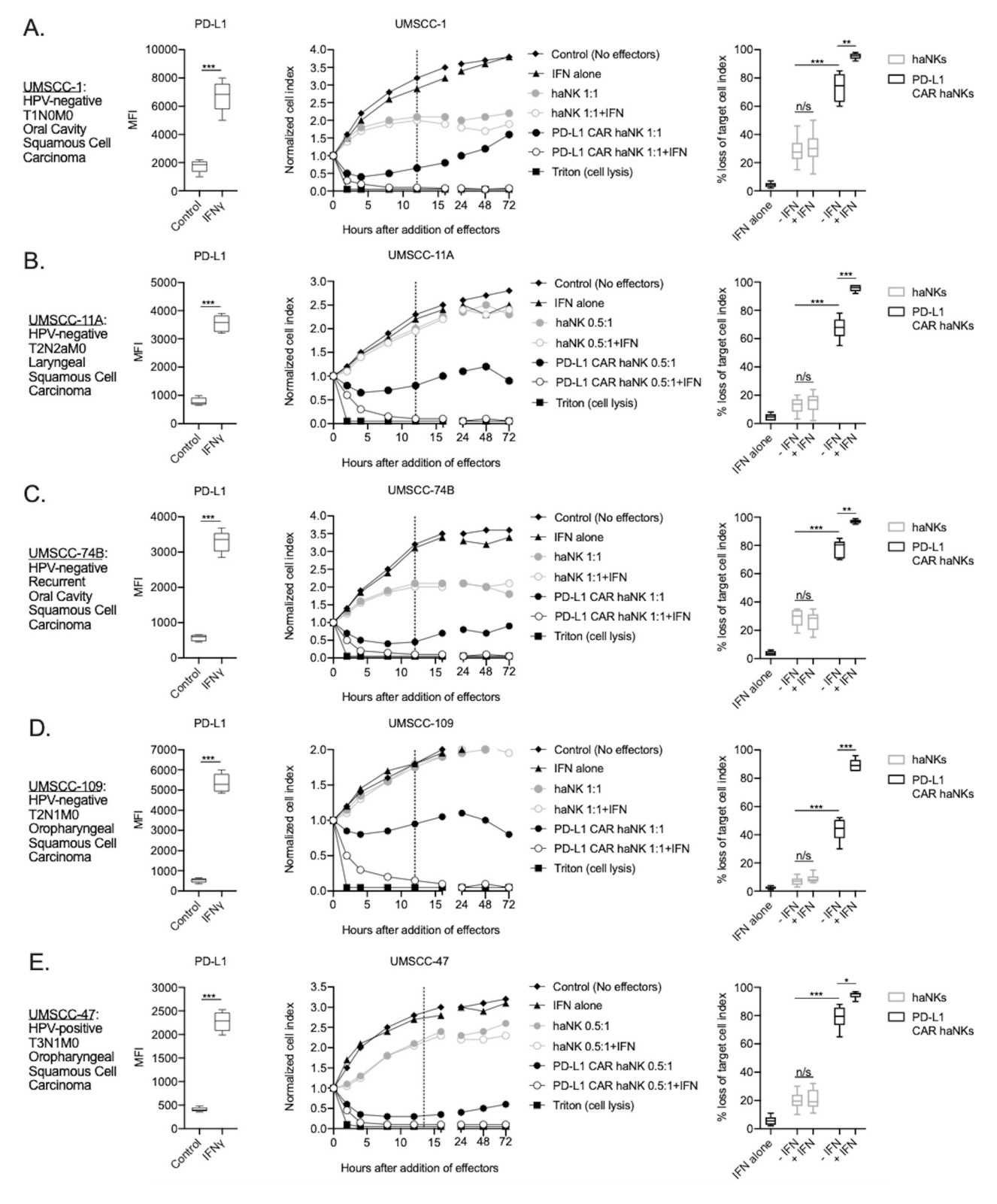

**Figure 1.** PD-L1 CAR haNKs demonstrated superior killing of human HNSCC targets compared to haNKs. Cell surface PD-L1 expression was measured by flow cytometry in the presence or absence of IFNγ (20 ng/mL for 24 hr, left panels) and killing of control or IFNγ pre-treated tumor cells (20 ng/mL for 24 hr) by haNKs or PD-L1 CAR haNKs was measured by impedance analysis for human UMSCC-1 (**A**), UMSCC-11A (**B**), UMSCC-74B (**C**), UMSCC-109 (**D**) and UMSCC-47 (**E**) HNSCC cells. Representative impedance curves are shown in the central panels, and plots of quantification of effector cell killing

*Figure 1 continued on next page*

*Figure 1 continued*

(% loss of cell index) at 12 hr are shown in the right panels. Cumulative data from at least three independent experiments for each cell line are shown. MFI, median fluorescent intensity. *, p<0.05; **, p<0.01; ***, p<0.001.

The online version of this article includes the following figure supplement(s) for figure 1:

**Figure supplement 1.** PD-L1 CAR haNKs express a PD-L1 CAR.

**Figure supplement 2.** UMSCC cells express variable degrees of HLA class I and MICA/B.

**Figure supplement 3.** PD-L1 CAR haNKs exhibit minimal exhaustion.

prior exposure to each cell line, respectively. Following efficient killing of target cells for a period of 4 hr, PD-L1 CAR haNKs retained their ability to kill target cells albeit at a reduced capacity compared to target-naïve PD-L1 CAR haNKs (*Figure 1—figure supplement 3*). Prior exposure to target cells partially reduces but does not abrogate the effector capacity of PD-L1 CAR haNKs.

The ability of PD-L1 CAR haNKs or haNKs to kill HPV negative murine HNSCC cells in the presence or absence of IFNγ pre-treatment was next determined. PD-L1 CAR haNKs killed IFNγ pretreated murine HNSCC cells in a dose-dependent fashion, with 70–80% cell killing at a 1:1 effector-to-target ratio (*Figure 2—figure supplement 1*). In the absence of IFNγ pre-treatment, baseline expression of PD-L1 on murine HNSCC cells was low, and PD-L1 CAR haNks and haNKs killed targets to a similar degree at 1:1 effector-to-target ratios (*Figure 2A&B*). IFNγ pre-treatment

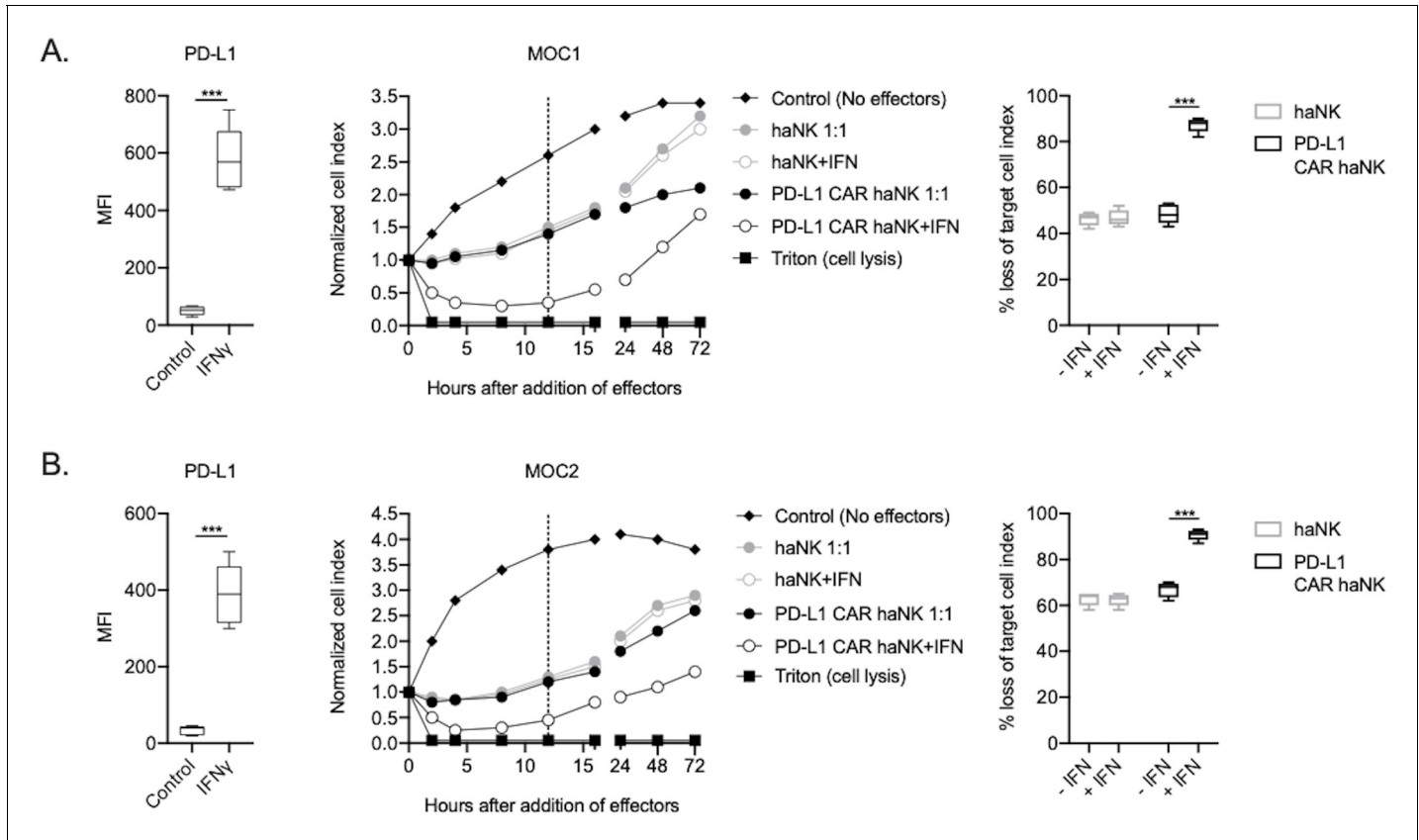

**Figure 2.** PD-L1 CAR haNKs demonstrated superior killing of murine HNSCC targets compared to haNKs. Cell surface PD-L1 expression was measured by flow cytometry in the presence or absence of IFNγ (20 ng/mL for 24 hr, left panels) and killing of control or IFNγ pre-treated tumor cells (20 ng/mL for 24 hr) by haNKs or PD-L1 CAR haNKs was measured by impedance analysis for murine MOC1 (**A**) or MOC2 (**B**) HNSCC cells. Representative impedance curves are shown in the central panels, and plots of quantification of effector cell killing (% loss of cell index) at 12 hr are shown in the right panels. Cumulative data from at least three independent experiments for each cell line are shown. MFI, median fluorescent intensity. ***, p<0.001.

The online version of this article includes the following figure supplement(s) for figure 2:

**Figure supplement 1.** PD-L1 CAR haNKs kill murine HNSCC cells in a dose-dependent fashion.

significantly increased expression of PD-L1 and killing by PD-L1 CAR haNKs, but no increase in killing by haNKs was observed. PD-L1 CAR haNK killing of IFN pre-treated murine target cells was less durable compared to human target cells, with a degree of rebound of target cell growth by 72 hr. These data revealed that PD-L1 CAR haNKs also kill murine HNSCC cells to a greater degree than haNKs, and that PD-L1 CAR haNK killing can be increased with IFNγ-induced PD-L1 expression.

To determine the relative contribution of direct and PD-L1 CAR-mediated killing of human and murine HNSCC targets by PD-L1 CAR haNKs, CRISPR/Cas9 gene editing was used to generate PD-L1 knockout cell variants of human UMSCC-1, human UMSCC-11A and murine MOC1 cells. Exposure of these PD-L1 knockout cells to IFNγ demonstrated a lack of baseline and IFNγ-inducible PD-L1 expression (compared to isotype control staining) but retained baseline and IFNγ-inducible HLA class I expression, demonstrating that overall IFNγ responses remained intact (*Figure 3—figure supplement 1*). Impedance analysis was next used to determine the ability of PD-L1 CAR haNKs to kill IFNγ pre-treated parental cells, PD-L1 knockout cells, or mixtures of these cells at different ratios (*Figure 3A–C*). Although killing of IFNγ pre-treated parental cells was nearly complete, killing of PD-L1 knockout cells ranged from 10–40%, suggesting that 10–40% of PD-L1 CAR haNK killing is due to direct NK cytotoxicity and independent of the PD-L1 CAR. Increased PD-L1 CAR haNK killing of targets correlated with increased ratios of parental-to-PD-L1 knockout cells. These data demonstrated that PD-L1 CAR haNKs utilize both direct and PD-L1 CAR-dependent killing of human and murine HNSCC cells.

The ability of PD-L1 CAR haNKs to kill murine HNSCC allows the in vivo study of efficacy and changes in immune correlatives in a syngeneic system. Immune-competent mice bearing established (100–200 mm$^3$) MOC1 tumors were treated with irradiated PD-L1 CAR haNKs, and the ability of these cells to induce tumor growth inhibition or changes in peripheral or tumor immune composition was measured. PD-L1 CAR haNKs ($1 \times 10^7$) were administered twice weekly given their short lifespan following irradiation. PD-L1 CAR haNK monotherapy induced tumor rejection in 30% of treated mice and tumor growth inhibition in a subset of remaining mice (*Figure 4A*). No weight loss, alterations in select blood chemistries, or other measurable forms of toxicity were observed with systemic PD-L1 CAR haNK treatment administered via intravenous or intraperitoneal routes (*Figure 4—figure supplement 1*). Tumor rejection was abrogated in mice bearing PD-L1 knockout MOC1 tumors (*Figure 4B*), but heterogeneous tumor growth inhibition was still observed. IFNγ is a primary driver of PD-L1 expression on MOC1 tumor cells (*Shah et al., 2016*). To determine if IFNγ is the major driver of PD-L1 being targeted by PD-L1 CAR haNKs and if CD8 T cells are the major source of IFNγ, treatments were repeated in the presence or absence of antibody depletions. IFNγ or CD8 depletion abrogated the anti-tumor effect of PD-L1 CAR haNKs (*Figure 4C*). Depletion of TNFα did not alter the anti-tumor effect of PD-L1 CAR haNKs. This data suggested that IFNγ produced by CD8 T cells was the major driver of tumor cell PD-L1 expression being targeted by PD-L1 CAR haNKs. Tumor cell-specific PD-L1 expression was significantly increased after PD-L1 CAR haNKs, and this increase was abrogated in the presence of CD8 depletion (*Figure 4D*). These data suggested that PD-L1 CAR haNKs can induce PD-L1-dependent tumor rejection in a syngeneic murine system, and that CD8 T cells are a primary source of IFNγ driving tumor cell PD-L1 expression in this model.

Circulating and tumor infiltrating immune cells can express high levels of PD-L1. To determine if treatment of MOC1 tumor bearing mice altered immune constituency, splenic and tumor immune subsets were measured by flow cytometry following PD-L1 CAR haNK administration (*Figure 5A*). In the periphery of tumor bearing mice, PD-L1 was uniformly expressed on macrophages and Ly6G$^{hi}$ neutrophilic myeloid cells to a greater degree compared to other cells types. Quantification of splenic immune cell fractions revealed that macrophages and neutrophilic myeloid cells endogenously expressing high levels of PD-L1 were decreased compared to control following treatment, whereas no changes were observed in other immune subsets (*Figure 5B&C*). In the tumor microenvironment, PD-L1 was expressed to a greater degree on macrophages, neutrophilic and monocytic myeloid cells compared to tumor cells or other immune subsets. Flow cytometric analysis 1 hr after treatment verified the presence of adoptively transferred PD-L1 CAR haNKs in the blood, spleens and tumors of MOC1 tumor bearing mice (*Figure 5—figure supplement 1*). Quantification of tumor immune infiltration revealed decreased macrophages, neutrophilic and monocytic myeloid cells, increased CD8 and CD4 T-lymphocytes, and no change in Tregs or NK cells following treatment (*Figure 5B&D*). To validate these findings in an ex vivo setting, peripheral or tumor leukocytes were isolated from mice bearing MOC1 tumors, co-incubated with PD-L1 CAR haNKs and flow cytometry

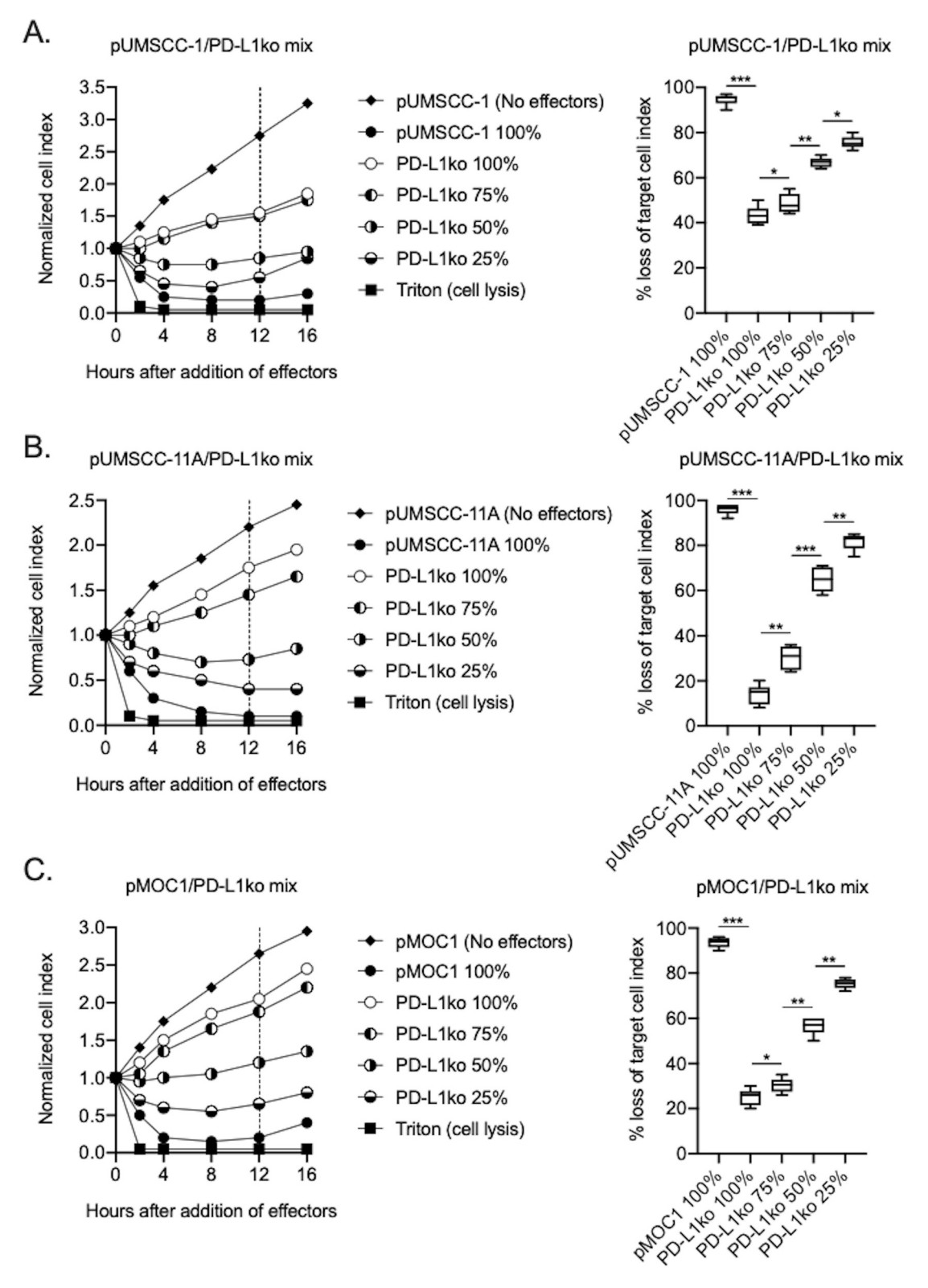

**Figure 3.** PD-L1 CAR haNKs demonstrated both direct and PD-L1 CAR-dependent killing. The ability of PD-L1 CAR haNK cells (1:1 effector:target ratio) to kill IFNγ pretreated parental and PD-L1 knockout UMSCC-1 (**A**), UMSCC-11A (**B**) and MOC1 (**C**) cells mixed at different ratios was measured by impedance analysis. Representative impedance plots are shown on the left, and plots of quantification of killing are shown on the right. Cumulative data from two independent experiments for each cell line are shown. *, p<0.05; **, p<0.01; ***, p<0.001.

*Figure 3 continued on next page*

*Figure 3 continued*

The online version of this article includes the following figure supplement(s) for figure 3:

**Figure supplement 1.** PD-L1 knockout cells maintained IFN responsiveness but lacked PD-L1 expression.

was used to determine immune constituency. Similar to results observed in vivo, immune cell subsets from the periphery and tumor that expressed comparatively greater levels of PD-L1 were reduced following 24 hr of co-incubation with PD-L1 CAR haNKs (*Figure 5—figure supplement 2*). Cumulatively, these data exhibited that PD-L1 CAR haNKs can mediate reduction of immune cells endogenously expressing high levels of PD-L1 in the periphery and tumor microenvironment of MOC1 tumor-bearing wild-type mice.

To determine if adoptive transfer of PD-L1 CAR haNKs could mediate tumor growth inhibition in a xenograft system lacking immunity, treatment was performed in severely immunodeficient NSG mice bearing UMSCC-1 human tumors. Tumor growth inhibition was observed, but no tumors were rejected (*Figure 6A*). Tumor growth inhibition following PD-L1 CAR haNK treatment was abrogated in NSG mice bearing PD-L1 knockout UMSCC-1 tumors (*Figure 6B*). Flow cytometry and immunofluorescence were used to verify that tumor cell-specific PD-L1 expression within UMSCC-1 tumors was increased following treatment with PD-L1 CAR haNKs (*Figure 6—figure supplement 1A&B*). PD-L1 CAR haNKs produce IFNγ at rest, and to a higher degree following target engagement (*Figure 6—figure supplement 1C*). These data suggested that the PD-L1 CAR haNKs may be the source of the IFNγ leading to increased expression of PD-L1 on the UMSCC-1 tumor cells. Cumulatively, these data demonstrated that PD-L1 CAR haNKs can also mediate PD-L1-dependent tumor growth inhibition in immunodeficient mice bearing human HNSCCs, possibly through the ability of PD-L1 CAR haNKs to induce IFNγ-dependent PD-L1 expression on tumor cells.

The ability of PD-L1 CAR haNKs to reduce the frequency of immune subsets endogenously expressing high levels of PD-L1 may be an important complementary mechanism of action to tumor cell killing. To explore whether this phenomenon can also be observed in patients with cancer, peripheral leukocytes from patients with advanced stage HPV negative HNSCC were co-incubated ex vivo with PD-L1 CAR haNK and changes in immune cell frequency were determined by flow cytometry (*Figure 7A*). In the peripheral blood of HNSCC patients, macrophages expressed the greatest levels of PD-L1, and CD14+ and CD15+ myeloid subsets expressed greater levels of PD-L1 compared to lymphoid or NK cells. PD-L1 high macrophages and CD14+/CD15+ myeloid cell subsets were significantly reduced following 24 hr of co-incubation with PD-L1 CAR haNKs (*Figure 7B&C*). These results validated that PD-L1 CAR haNKs possess the ability to reduce the cell frequency of leukocytes endogenously expressing high PD-L1 from patients with HNSCC.

## Discussion

As our understanding of the limitations of T cell-based immunotherapy increases, it is evident that alternative immunotherapy approaches to be used in lieu of or in combination with immune checkpoint blockade or T cell cellular therapies are needed. As NK cells detect and eliminate tumor cells via MHC- and antigen-independent mechanisms, NK cell-based immunotherapy may overcome some mechanisms of resistance to T cell-based immunotherapy, including selection of tumor subclones lacking antigen or harboring antigen processing/presentation defects (reviewed in *Greene et al., 2019*). The presence of genomic alterations in one or more of these antigen processing and presentation pathways predicts response to all forms of T cell-based immunotherapy including immune checkpoint blockade and adoptive transfer cell therapies (*Doran et al., 2019*; *Shin et al., 2017*; *Zaretsky et al., 2016*; *Gao et al., 2016*). Most existing data have explored whole tumor biopsies for the presence of one or more of these genomic alterations and correlation to response to immunotherapy. However, high degrees in tumor heterogeneity within epithelial malignancies result in multiple subclones of tumor cells (*Morris et al., 2016*). The number of tumor subclones present within an individual tumor that harbors one or more of these genomic alterations is unknown. This may be an important predictor of whom would most benefit from NK cell-based immunotherapy as an alternative or adjunct to T cell-based immunotherapy.

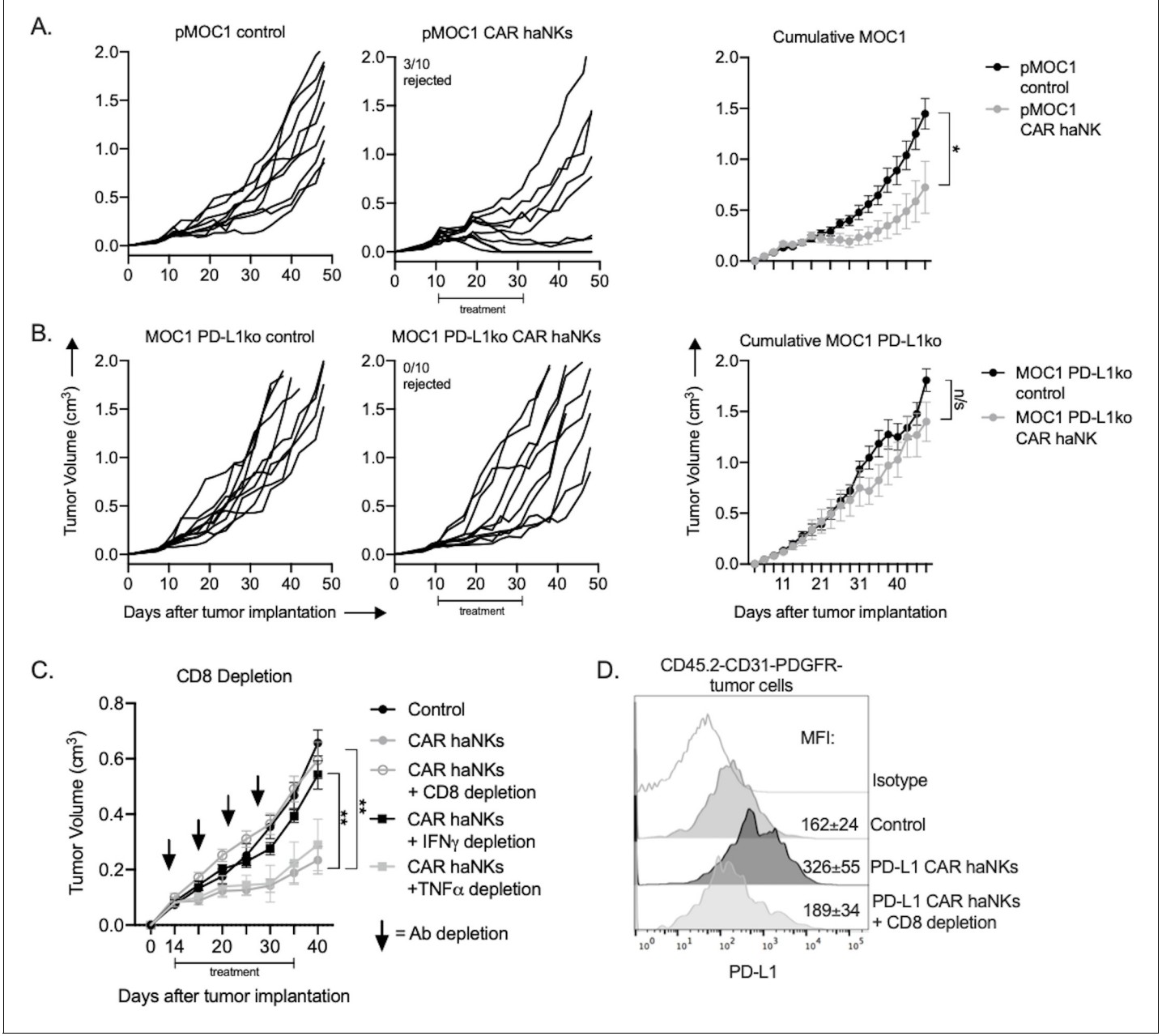

**Figure 4.** PD-L1 CAR haNKs induced PD-L1-dependent rejection or tumor growth inhibition in wild-type mice bearing MOC1 tumors. Wild-type C57BL/6 mice bearing parental (A) or PD-L1 knockout (B) MOC1 tumors were treated with PD-L1 CAR haNKs ($1 \times 10^7$ cells IP, beginning day 10, twice weekly for six doses, $n$ = 10 mice/group) or 1xPBS control and primary tumors were followed for growth. Individual tumor growth curves shown on the left, with summary growth curves shown on the right. The number of mice that rejected established tumors is inset. C, wild-type C57BL/6 mice bearing parental MOC1 tumors were treated with PD-L1 CAR haNKs ($1 \times 10^7$ cells IP, beginning day 14, twice weekly for six doses, $n$ = 10 mice/group) or 1xPBS control with and without antibody-based depletion (CD8 clone YTS 169.4, IFNγ clone XMG1.2, TNFα clone XT3.11, each 200 μg IP twice weekly starting on day 13, $n$ = 5–10 mice/group) and primary tumors were followed for growth. Summary growth curve shown. D, one day after the final CD8 depletion and two days after the final PD-L1 CAR haNK treatment, some tumors ($n$ = 4) were assessed for tumor cell-specific PD-L1 expression by flow cytometry. Representative histograms shown with median fluorescent intensity (MFI) inset. Significance of summary growth curves was determined by 2-way ANOVA of repeat measures with the treatment condition as the variable. Results from one of three independent experiments with similar results are shown. *, $p < 0.05$.

The online version of this article includes the following figure supplement(s) for figure 4:

**Figure supplement 1.** PD-L1 CAR haNKs exhibit no measurable toxicity.

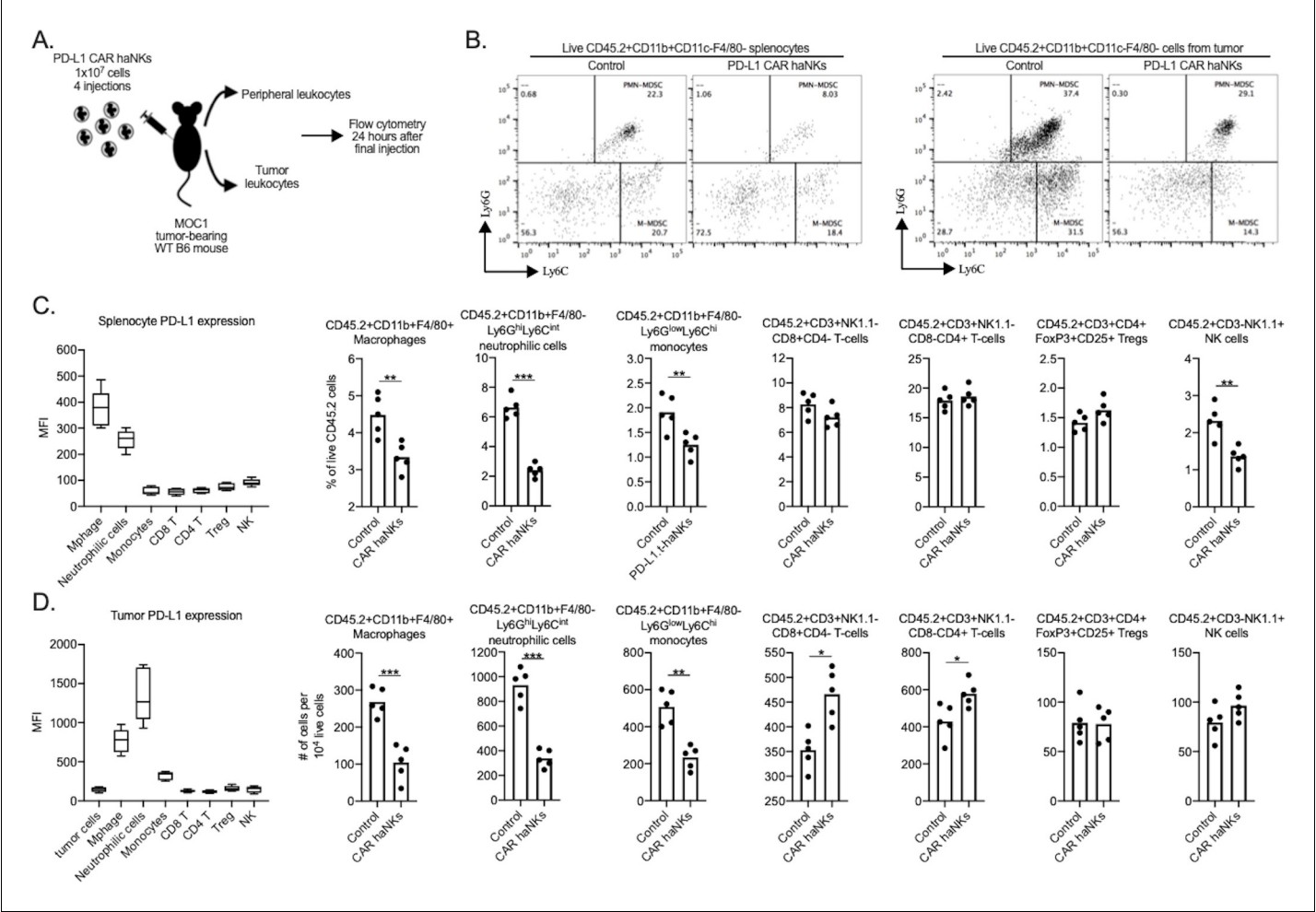

**Figure 5.** PD-L1 CAR haNK treatment remodels the tumor microenvironment immune composition in wild-type mice bearing. MOC1 tumors Wild-type C57BL/6 mice bearing parental MOC1 tumors were treated with four PD-L1 CAR haNK administrations (1 × 10$^7$ cells IP, each four days apart) and 24 hr after the last treatment, spleens and tumors were harvested and assessed for immune composition via flow cytometry (schema shown in **A**). (**B**) representative dot plots of live CD11c-F4/80- myeloid cells from the spleen (left panels) or tumor (right panels) are shown. Baseline cell surface PD-L1 expression on immune cell subsets along with changes in immune cell composition following PD-L1 CAR haNK treatment are shown from the spleen (**C**) and tumor (**D**) compartments (n = 5 spleens or tumors/group). Cell surface markers used to identify immune cell subsets are shown. *, p<0.05; **, p<0.01; ***, p<0.001.

The online version of this article includes the following figure supplement(s) for figure 5:

**Figure supplement 1.** Kinetics of PD-L1 CAR haNKs in bloods, spleens and tumors.

**Figure supplement 2.** PD-L1 CAR haNKs deplete PD-L1 high myeloid cells from the spleen and tumors of MOC1 tumor bearing mice.

The development of the NK-92 cell line from a patient with Non-Hodgkin's lymphoma allowed a continuously cultured effector NK cell line to be used as an 'off the shelf' cell therapy product (*Gong et al., 1994*). Repeat infusions of high doses of NK-92 cells had a promising safety profile and induced clinical benefit in some patients (*Williams et al., 2017*; *Tonn et al., 2013*). NK-92 cells lack most inhibitory killer immunoglobulin-like receptors (KIRs) but require exogenous IL-2 for proliferation in culture (*Yan et al., 1998*; *Maki et al., 2001*). To overcome this requirement, haNKs, or high-affinity NK cells, are NK-92 cells engineered to express endoplasmic reticulum-retained IL-2(11). haNKs have demonstrated the ability to efficiently kill carcinoma cells at low effector-to-target ratios (*Friedman et al., 2019*; *Friedman et al., 2018*). Irradiation of haNKs and haNK-derived cell products ensures short life span of these cells and allows for repeat administration with no risk of cellular engraftment and low risk of cytopenias in treated hosts (*Seery et al., 2019*; *Brudno and Kochenderfer, 2016*).

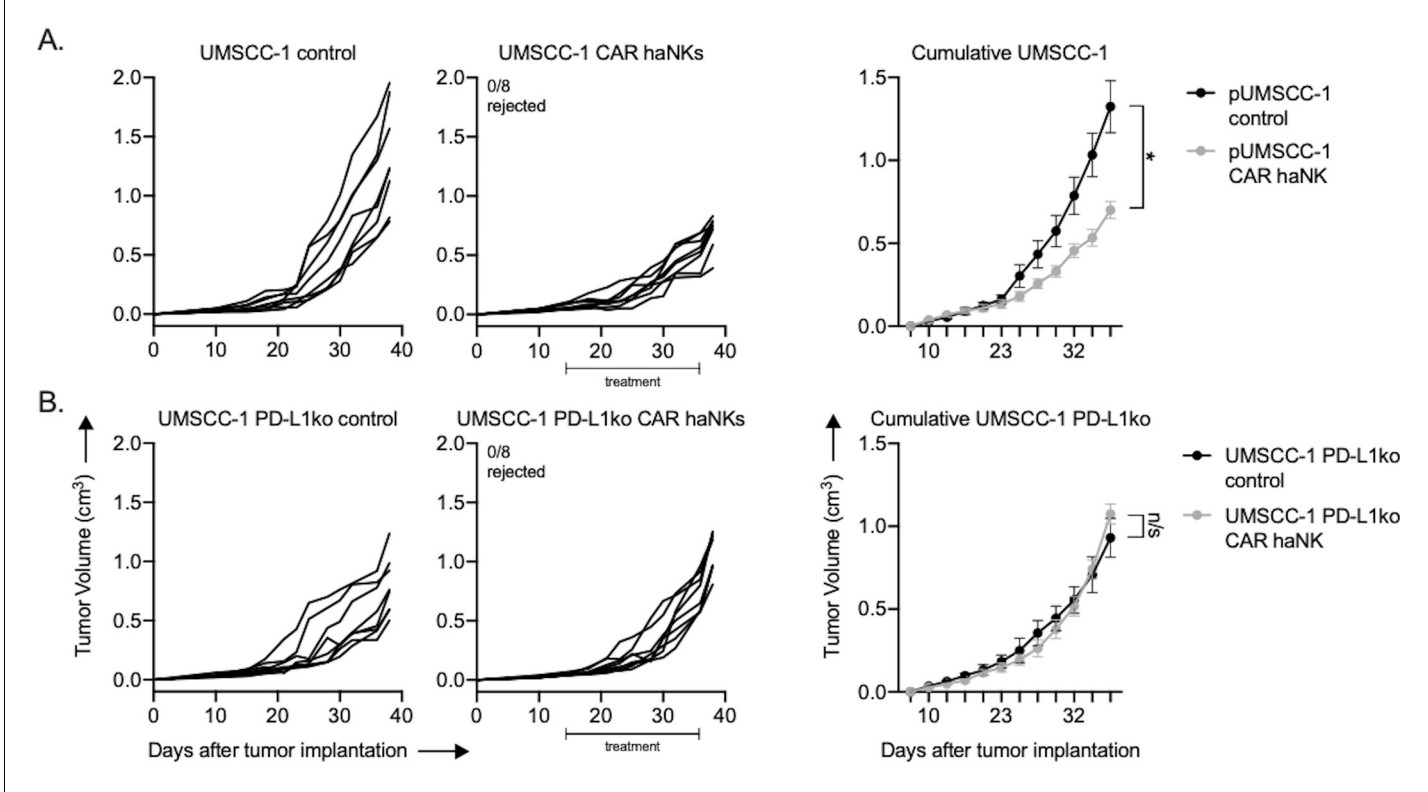

**Figure 6.** PD-L1 CAR haNKs induced PD-L1-dependent tumor growth inhibition in NSG mice bearing UMSCC-1 tumors. NOD-*scid* IL2Rgamma^null (NSG) mice bearing parental (A) or PD-L1 knockout (B) UMSCC-1 tumors were treated with PD-L1 CAR haNKs ($1 \times 10^7$ cells IP, beginning day 14, twice weekly for six doses, $n$ = 10 mice/group) or 1xPBS control and primary tumors were followed for growth. Individual tumor growth curves shown on the left, with summary growth curves shown on the right. The number of mice that rejected established tumors is inset. Significance was determined by 2-way ANOVA of repeat measures with the treatment condition as the variable. Results from one of two independent experiments with similar results are shown. *, $p<0.05$.

The online version of this article includes the following figure supplement(s) for figure 6:

**Figure supplement 1.** UMSCC-1 tumor cells express greater levels of PD-L1 after a treatment with PD-L1 CAR haNKs.

Here, we demonstrated the in vitro and in vivo efficacy of haNKs modified to express a PD-L1 CAR. Although PD-L1 CAR haNKs mediated cytotoxicity through both direct and PD-L1 CAR-dependent mechanisms, killing of targets was primary PD-L1-dependent as tumor cell PD-L1 knockout reduced in vitro killing capacity by 60–90% and abrogated in vivo tumor rejection or growth inhibition. This suggests that PD-L1 CAR haNK monotherapy may be most effective in patients harboring PD-L1+ tumors with evidence of genomic alterations or expression defects predicted to lead to insensitivity to T cell detection or elimination. Greater than 80% of patients with HNSCC have a combined positive score of >1 indicative of PD-L1 positivity (*Chow et al., 2016*), but significant heterogeneity of PD-L1 expression exists within individual tumors (*Rasmussen et al., 2019*). PD-L1-dependent and independent mechanisms of killing by PD-L1 CAR haNKs are likely to be required for maximum treatment efficacy.

Our data demonstrated that mouse and human peripheral and tumor infiltrating myeloid cells express greater levels of PD-L1 compared to other immune cells populations, and that PD-L1 CAR haNKs can reduce accumulation of these PD-L1 high myeloid cell populations. The selective elimination of PD-L1 high myeloid cells by PD-L1 CAR haNKs may be an important complimentary mechanism of action to direct or PD-L1-dependent tumor cell killing. Macrophages, neutrophilic myeloid cells and monocytic myeloid cells are immunosuppressive through multiple independent mechanisms, especially upon trafficking into the tumor microenvironment, and limit effective baseline and immunotherapy-induced T cell and NK cell anti-tumor immunity (*Condamine et al., 2016*; *Li et al., 2009*; *Noguchi et al., 2017*; *Arlauckas et al., 2017*). PD-L1 expression on myeloid cells is sufficient

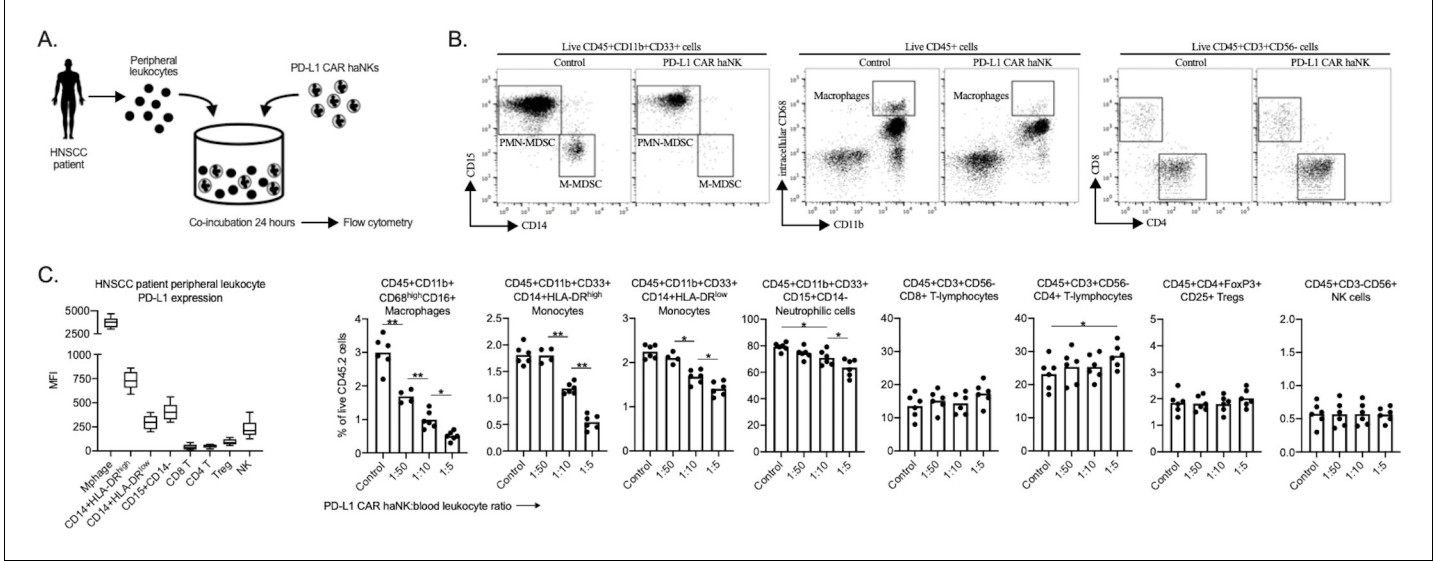

**Figure 7.** PD-L1 CAR haNKs deplete PD-L1 high myeloid cells from the peripheral blood of HNSCC patients PD-L1. CAR haNKs were co-incubated for 24 hr at different effector:target ratios with HNSCC patient peripheral blood leukocytes (n = 5 patients) and changes in immune composition were assessed via flow cytometry (schema shown in A). (B) Representative dot plots of myeloid and lymphoid cell subsets are shown. (C) Baseline cell surface PD-L1 expression on immune cell subsets along with changes in immune cell composition following PD-L1 CAR haNK co-incubation are shown. *, p<0.05; **, p<0.01.

and may even be required for immunosuppression mediated by PD-pathway signaling (*Noguchi et al., 2017*; *Lau et al., 2017*). We and others have demonstrated that elimination of myeloid cells or inhibition of their trafficking into tumors has little effect as a monotherapy but enhances T cell cell-based immunotherapy when used in combination (*Sun et al., 2019*; *Highfill et al., 2014*; *Steele et al., 2016*). A natural extension of the work presented here is to combine PD-L1 CAR haNKs with different T cell or NK cell-based immunotherapies to explore if reduction in myeloid cell populations can enhance responses to other immunotherapies. This treatment approach deserves further study and is ongoing in our and other laboratories.

The decision to introduce a PD-L1 CAR into haNKs was made based upon the known immunosuppressive mechanism of PD-L1 and the possibility that both tumor and PD-L1 positive immune cell subsets would be targeted. CAR expression in haNKs may be advantageous over introduction of CARs into autologous T cells as haNK cells lack expression of a native TCR that could mediate an allogeneic immune reaction such as graft-vs-host disease (*Olson et al., 2010*; *Miller et al., 2005*). This suggests that haNK-based CAR immunotherapy may be useful across populations of patients with cancer and not just in individual patients (reviewed in *Mehta and Rezvani, 2018*).

A major limitation of the conclusions of our data was the use of subcutaneous flank xenograft and syngeneic models that do not metastasize. The use of an orthotopic metastatic model transplanted into the oral cavity region could provide valuable insight into the ability of PD-L1 CAR haNKs to prevent or delay the development or the progression of metastases. This requires further study. Another limitation of our study is our technical inability to detect PD-L1 CAR haNKs in resected malignant tissues by immunofluorescence. This data could provide valuable insight into the localization of adoptively transferred cells that trafficking to the tumor. Work to develop validated assays to determine tumor special localization of NK cell products is ongoing.

Based upon our data along with the data of others, CAR targeting of NK cells appears to enhance their killing efficacy (*Ingegnere et al., 2019*; *Liu et al., 2018*). To date, most progress has been made in the treatment of hematologic malignancy, with cord blood NK cells engineered to express a CD19-targeting CAR demonstrating high response rates in patients with relapsed CD19+ leukemia or lymphoma (*Liu et al., 2020*). CARs targeting other molecules constitutively expressed on tumor cells from solid epithelial malignancies such as EGFR and HER2 may also be useful and deserve further study (*Chen et al., 2016*; *Nowakowska et al., 2018*). With all CAR-based therapies targeting surface molecules that have wide expression patterns in patients, careful phase I monotherapy study

is required to determine if limiting on-target but off-tumor toxicities exist. Targeting PD-L1 may allow selective targeting of tumor cells and immunosuppressive, PD-L1 high immune cells, and side effect profiles would be predicted to be similar to profiles observed with PD-1/L1 immune check-point blockade. The data presented here provides the pre-clinical rationale needed to perform early phase clinical trials to explore the safety profile of PD-L1 CAR haNKs in patients with relapsed or metastatic cancer.

# Materials and methods

## Key resources table

| Reagent type (species) or resource | Designation | Source or reference | Identifiers | Additional information |
|---|---|---|---|---|
| Cell line (human) | UM-SCC-1 | U of Michigan | | Available from EMDMillipore |
| Cell line (human) | UM-SCC-11A | U of Michigan | | |
| Cell line (human) | UM-SCC-74B | U of Michigan | | Available from EMDMillipore |
| Cell line (human) | UM-SCC-109 | U of Michigan | | |
| Cell line (human) | UM-SCC-47 | U of Michigan | | Available from EMDMillipore |
| Cell line (mouse) | MOC1 | WashU St. Louis | | Available from Kerafast |
| Cell line (mouse) | MOC2 | WashU St. Louis | | Available from Kerafast |
| Cell line (human) | UM-SCC-1 PD-L1 ko | This paper | | Knockout by Synthego, cells available from senior author |
| Cell line (human) | UM-SCC-11A PD-L1 ko | This paper | | Knockout by Synthego, cells available from senior author |
| Cell line (mouse) | MOC1 PD-L1 ko | This paper | Sc-425636 | Santa Cruz knockout reagents, cells available from senior author |
| NK cell therapy | haNKs | NantKWest | | Effector cells |
| NK cell therapy | PD-L1 CAR haNKs | NantKWest | | Effector cells |
| Recombinant protein | Human IFNγ | Peprotech | 300–02 | In vitro stim (20 ng/mL) |
| Recombinant protein | Murine IFNγ | Biolegend | 575304 | In vitro stim (20 ng/mL) |
| Antibody | CD8 | BioXCell | YTS 169.4 | In vivo depletion (200 µg/injection) |
| Antibody | IFNγ | BioXCell | XMG1.2 | In vivo depletion (200 µg/injection) |
| Antibody | TNFα | BioXCell | XT3.11 | In vivo depletion (200 µg/injection) |
| Commercial assay or kit | IFNγ ELISA | R and D Systems | DIF50 | |
| Antibody | PD-L1 | Cell Signaling | E1L3 | IF (1:1000) |

## Tumor cells and culture

A panel of four human papillomavirus (HPV) negative (UM-SCC-1, UM-SCC-11A, UM-SCC-74B and UM-SCC-109) and one HPV positive (UM-SCC-47) HNSCC cell lines were obtained from Drs. T. E. Carey, M. E. Prince, and C. R. Bradford at the University of Michigan (Ann Arbor, MI) and cultured as described (*Brenner et al., 2010*). Original stocks of genomically characterized (*Onken et al., 2014*) parental mouse oral cancer 1 (MOC1) and MOC2 cells were obtained from R. Uppaluri at Washington University (St. Louis, MO) and cultured as described (*Judd et al., 2012*). Cell lines were serially verified to be free of mycoplasma and other pathogens and used at low (<30) passage number. Cells

were harvested for experiments using 0.25% trypsin. In some experiments, cells were exposed to human (PeproTech) or murine (Biolegend) recombinant IFNγ.

NK cell therapy products haNK cells (*Jochems et al., 2016*) and haNK cells transduced with a second-generation CAR targeting PD-L1 produced through proprietary transduction and cell culture techniques were obtained from NantKWest through a Cooperative Research and Development Agreement. Cells were irradiated (15 Gy) prior to cryopreservation and shipment from NantKWest to the NIH for experimental use.

## PD-L1 knockout cells

PD-L1 knockout was achieved with CRISPR/Cas9 genome editing without antibiotic selection using commercially available plasmids for human (Synthego) and murine (Santa Cruz) cells. Following CRISPR/Cas9 genome editing, cells were exposed to IFNγ (20 ng/mL for 24 hr) and sorted to >99% knockout purity (PD-L1 negative) using fluorescence-activated cell sorting (BD Aria III sorter).

## In vivo experiments

Wild-type C57BL/6 mice were purchased from Taconic and NOD-*scid* IL2Rgamma[null] (NSG) mice were purchased from Jackson Laboratories. Mice were housed in a pathogen-free environment and all experiments were performed under an Animal care and Use Committee approved protocol. Syngeneic murine or xenograft human tumors were established by subcutaneous flank injection of tumor cells in Matrigel (Trevigen, 30% by volume). Mice were assessed for tumor growth three times weekly and tumor volume was calculated as: (length$^2$ x width)/2. In some experiments, antibody-based depletion was performed via intraperitoneal (IP) injection of a CD8 (clone YTS 169.4, Bio-XCell), IFNγ (clone XMG1.2) or TNFα (clone XT3.11) mAb, 200 μg each twice weekly starting 13 days following tumor implantation. Murine blood chemistries were performed by the National Institutes of Health Department of Laboratory Medicine.

## Impedance analysis

Real-time assessment of alterations in cell viability upon exposure to effector cells was measured via impedance analysis using the xCELLigence RTCA platform as described (*Sun et al., 2018*). For all experiments, target cells were plated (1 × 10$^4$ cells/well) and allowed to adhere and gain impedance overnight in exponential growth phase before the addition of effectors. For each experiment, the effector-to-target ratio detailed in the figure legends is based upon the number of initially plated target cells. Triton X-100 (0.2%) was used as a positive control for complete cell lysis. Percent loss of cell index was calculated as: 1 - (experimental cell index/control cell index) for a given timepoint.

## Flow cytometry

PD-L1 CAR haNK cells were assessed for PD-L1 CAR expression via staining with biotinylated recombinant human PD-L1 (ACROBiosystems) followed by staining with a fluorophore conjugated to streptavidin. Cultured tumor cells harvested for flow analysis were exposed to CD16/32 (FcR block) antibodies for 10 min prior to staining with primary conjugated antibodies for 30 min. For some experiments, spleen and tumors were harvested from mice and processed into single cell suspensions for flow analysis. Only fresh tissues processed as described (*Clavijo et al., 2017*) were analyzed. Following FcR block, primary conjugated antibodies were applied for 45 min. Anti-human PD-L1 (clone 29E.203), MICA/B (6D4), HLA-A/B/C (W6/32), CD45 (H130), CD31 (WM59), CD140a/PDGFR (aR1), CD11b (ICRF44), CD68 (Y1/82A), CD16 (3G8), CD33 (WM53), CD14 (M5E2), CD15 (W6D3), HLA-DR (L243), CD3 (SK7), CD4 (OKT4), CD8 (SK1), FoxP3 (206D), CD11c (3.9), CD25 (M-A251) and anti-mouse PD-L1 (10F.9G2), CD45.2 (104), CD31 (390), CD140a/PDGFR (APA5), CD11b (M1/70), F4/80 (Bm8), Ly6G (1A8), Ly6C (HK1.4), CD3 (17A2), NK1.1 (PK136), CD4 (GK1.5), CD8 (53–6.7), FoxP3 (FJK-16s), CD11c (N418), CD25 (PC61.5) antibodies were purchased from Biolegend, Thermo or BD Biosciences. Isotype control antibodies (Biolegend) and fluorescence-minus-one approaches were used to ensure staining specificity. For some experiments, cells were fixed and permeabilized for intranuclear staining using the FoxP3 transcription factor staining buffer set (eBioscience) per manufacturer recommendations. Dead cells were excluded using non-fixable (Sytox, Thermo) or fixable (Zombie, BioLegend) viability dyes. All analyses were performed on a BD Fortessa analyzer running FACSDiva software and interpreted using FlowJo (vX10.0.7r2).

### IFNγ ELISA

Human IFNγ ELISA kit was purchased from R and D Systems and used per manufacturer protocol.

### Cancer patient samples

Whole peripheral blood from patients with advanced stage HPV-negative head and neck cancer was obtained via venipuncture in green top heparinized tubes under an IRB-approved biospecimen procurement protocol (NIDCD 18-DC-0051, NCT03429036). Leukocytes were enriched using a 40/80% Percoll density gradient (cells suspended in upper 40% layer with 80% underlay, 325xg for 23 min at room temperature with slow centrifuge brake and acceleration).

### Multiplex immunofluorescence staining and multispectral imaging

We stained 5-μm-thick formalin-fixed paraffin-embedded sections from human xenograft tumors (UMSCC-1) using Opal multiplex kits (PerkinElmer/Akoya), for a panel of DAPI, and PD-L1 (clone E1L3) internally validated on a fully automated platform (Leica Bond RX). Multispectral images were acquired using Polaris System (PerkinElmer/Akoya).

### Statistics

Tests of significance between pairs of data are reported as p-values, derived using a student's t-test with a two-tailed distribution and calculated at 95% confidence. Comparison of multiple sets of data was achieved with analysis of variance (ANOVA) with Tukey's multiple comparisons. Differences in summary tumor growth curves were determined via 2-way ANOVA of repeat measures with the treatment condition as the variable All error bars indicate standard deviation. Statistical significance was set to $p < 0.05$. All analysis was performed using GraphPad Prism v7.

## Acknowledgements

The authors thank Dr. Chris Silvin for flow cytometry expertise, and Drs. Nyall London and Harris Floudas for their critical review of this work.

## Additional information

#### Competing interests

John H Lee, Patrick Soon-Shiong: Employed by NantKWest. The other authors declare that no competing interests exist.

#### Funding

| Funder | Grant reference number | Author |
| --- | --- | --- |
| National Institutes of Health | ZIA-DC00008 | Yvette Robbins<br>Sarah Greene<br>Jay Friedman<br>Paul E Clavijo<br>Carter Van Waes<br>Clint T Allen |

The funders had no role in study design, data collection and interpretation, or the decision to submit the work for publication.

#### Author contributions

Yvette Robbins, Sarah Greene, Data curation, Formal analysis, Investigation, Methodology, Writing - original draft, Writing - review and editing; Jay Friedman, Paul E Clavijo, Kellsye P Fabian, Michelle R Padget, Houssein Abdul Sater, Data curation, Formal analysis, Investigation, Methodology, Writing - review and editing; Carter Van Waes, Formal analysis, Supervision, Project administration, Writing - review and editing; John H Lee, Conceptualization, Resources, Funding acquisition, Investigation, Writing - review and editing; Patrick Soon-Shiong, Resources, Formal analysis, Funding acquisition, Writing - review and editing; James Gulley, Conceptualization, Resources, Formal analysis, Funding

acquisition, Writing - review and editing; Jeffrey Schlom, Conceptualization, Resources, Formal analysis, Funding acquisition, Investigation, Methodology, Writing - review and editing; James W Hodge, Conceptualization, Resources, Data curation, Formal analysis, Investigation, Methodology, Writing - review and editing; Clint T Allen, Conceptualization, Resources, Data curation, Software, Formal analysis, Supervision, Validation, Investigation, Visualization, Methodology, Writing - original draft, Project administration, Writing - review and editing

## Author ORCIDs
James Gulley http://orcid.org/0000-0002-6569-2912
Clint T Allen https://orcid.org/0000-0001-6586-5804

## Ethics

Human subjects: All blood was collected following informed patient consent onto an IRB-approved Biospecimen Procurement Protocol (National institutes of Health Protocol number 18-DC-0051, NCT number 03429036). This protocol follows guidelines established by the US Common Rule.

Animal experimentation: This study was performed in strict accordance with the recommendations in the Guide for the Care and Use of Laboratory Animals of the National Institutes of Health. All of the animals were handled according to approved institutional animal care and use committee (IACUC) protocols (number 1364-18) of the National Institutes of Health. Every effort was made to minimize suffering.

## Decision letter and Author response
Decision letter https://doi.org/10.7554/eLife.54854.sa1
Author response https://doi.org/10.7554/eLife.54854.sa2

# Additional files

## Supplementary files
• Transparent reporting form

## Data availability

All data generated or analysed during this study are included in the manuscript and supporting files. Source data files has been uploaded into Dryad in the form of GraphPad Prism file. One Prism file contains the raw data for all of the figures.

The following dataset was generated:

| Author(s) | Year | Dataset title | Dataset URL | Database and Identifier |
|---|---|---|---|---|
| Robbins Y, Greene S, Friedman J, Clavijo PE, Van Waes C, Fabian KP, Padget MR, Abdul Sater H, Lee JH, Soon-Shiong P, Gulley J, Schlom J, Hodge JW, Allen CT | 2020 | Data from: Tumor control via targeting PD-L1 with chimeric antigen receptor modified NK cells | https://doi.org/10.5061/dryad.sf7m0cg3k | Dryad Digital Repository, 10.5061/dryad.sf7m0cg3k |

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
