## [Decision Letter]

Thank you for submitting your article "Tumor control via targeting PD-L1 with chimeric antigen receptor modified NK cells" for consideration by *eLife*. Your article has been reviewed by three peer reviewers, one of whom is a member of our Board of Reviewing Editors, and the evaluation has been overseen by Satyajit Rath as the Senior Editor. The following individual involved in review of your submission has agreed to reveal their identity: Matti Korhonen (Reviewer #2).

The reviewers have discussed the reviews with one another and the Reviewing Editor has drafted this decision to help you prepare a revised submission.

Summary:

This paper by Robbins et al. shows an interesting and novel approach of NK-based CAR T cell therapy targeting PDL1. The authors hypothesize that the PD-L1 CAR haNK cells will attack both tumor cells expressing PD-L1, as well as the patient's own (mainly myeloid) PD-L1 expressing immune suppressor cells. In fact, the authors nicely show that irradiated haNKs CAR-PDL1 cells are able to induced cytotoxicity against human papillomavirus-positive and negative human and murine HNSCC cells. Moreover, they have proved that NK92 cells preserve their ability to kill tumor cells even after the CAR expression. in vivo data also have been generated showing a greater activity of haNKs CAR-PDL1 respect to haNKs.

Although novel and interesting the manuscript has raised different concerns and it would be important for the authors to address these.

Essential revisions:

1) Authors have not described the CAR construct, nor references have been provided. The authors state that they used a 'second generation CAR'. The structure of the CAR, however is of utmost importance for readers to be able to evaluate the work. The authors need to describe the structure in more detail (add also schematic diagram), including the nature of the costimulatory, transmembrane and spacer domains. They also need to describe the structure and source (e.g. which antibody clone) of the PD-L1 binding domain of the CAR. Also the viral vector and transduction process as well as the cell culture conditions need to be described.

2) In the first in vivo experiment, in particular the authors show that in the WT mice bearing MOC1 tumors, the CD8 depletion abrogated completely tumor growth inhibition. They attribute this effect to CD8 T cells being the primary source of INF-γ driving the tumor cell PD-L1 expression in the model. This observation somehow conflicts with results in Figure 2 showing 50% killing of MOC1 cells without INF. In my opinion it could also be that the PD-L1 CAR haNK cells are binding PD-L1 but not killing the cells yet the blocking of PD-L1 is allowing CD8 T cells to kill the tumor cells, explaining why when CD8 T cells were depleted tumor inhibition was not seen anymore. To claim that the reason tumor inhibition was abrogated after CD8 depletion was due to INFγ, the authors should re-do the experiment with a group of mice that are CD8 depleted and treated with PD-L1 CAR haNK and intratumoral INFγ. This would then show that the PD-L1 CAR haNK cells are able to kill in vivo and not acting similarly to a blocking antibody.

3) Various autoimmune symptoms are a major side effect of immune checkpoint inhibitors in the clinic. The availability of an immunocompetent mouse model may allow the interrogation of this phenomenon. Did these mice display signs of autoimmunity? In this paper there has not been experiments commenting on the toxicity of the therapy and this somehow gives a truncated story. Some toxicity studies should be added to complete the story or alternatively, the authors may present the rationale why such symptoms are not expected to arise in this experimental setting, perhaps due to the short time span of the experiment.

4) The kinetics of PD-L1 haNK cells in the immunocompetent mouse system are important to describe. The authors demonstrate that PD-L1 CAR haNK cells infiltrate MOC1 tumors at 1 hour after cell infusion but that they disappear by 24 hours. What is their kinetics in the blood, and in the spleen?

5) Strong limitation of the murine models is represented by the fact that the authors have performed a subcutaneous flank injection of tumor cells, and not have provided activity in a orthotopic tumor model for HNSCC. This is a special requirement for HNSCC since the relevance of lymph node metastasis in this tumor model. An orthotopic tumor model could answer if haNKs CAR are able to migrate to the metastic and primary tumor site. Moreover, since PDL1 is expressed in several healthy tissues, a systemic iv infusion of the haNKs CAR-PDL1 need to be performed, to verify whether on-target off-tumor toxicity could limit the clinical application of the approach.

6) Impedance analysis is suitable only for short term ability of haNKs CAR-PDL1 to control the tumor. Long-term culture of at least 72hrs is required to verify antitumor control, even considering that haNKs have been irradiated.

7) In either Introduction or Discussion, authors have not been discussed recent and relevant paper in NK CAR field (e.g. Liu et al., 2018 and 2020; Ingegnere et al., 2019).

8) Figure 3—figure supplement 1 shows the expression pattern of PD-L1 knockout human UMSCC-1, human UMSCC-11A and murine MOC1 cells. The MFI is significantly reduced in the PD-L1 knockout cells, however the expression seems not completely negative. The authors should show Isotype controls to clarify whether is only a matter of setting of the flow-analysis or if after the gene editing, the PDL1 has a dim expression that CAR.PDL1 is unable to recognize.

9) Figure 6—figure supplement 1 shows that PDL1 is upregulated only marginally in the tumor section. Maybe this is reflecting haNKs CAR-PDL1 localization. It would be interesting if the Authors could integrate the staining of NK cells in combination with PDL1.

---

## [Author Response]

Essential revisions:1) Authors have not described the CAR construct, nor references have been provided. The authors state that they used a 'second generation CAR'. The structure of the CAR, however is of utmost importance for readers to be able to evaluate the work. The authors need to describe the structure in more detail (add also schematic diagram), including the nature of the costimulatory, transmembrane and spacer domains. They also need to describe the structure and source (e.g. which antibody clone) of the PD-L1 binding domain of the CAR. Also the viral vector and transduction process as well as the cell culture conditions need to be described.

We apologize for omitting this information initially, and we agree this is important information to disclose for readers. We have added a schematic of the structure of the CAR to a new Figure 1—figure supplement 1A. Additionally, we have added the following text to the Results section: “This second-generation CAR includes a single-chain variable fragment (scFv) derived from NANT-601, and IgG1 mAb targeting PD-L1, along with a CD8 hinge, a CD28 transmembrane domain and an intracellular FcεR1γ signaling domain (diagramed in Figure 1—figure supplement 1A).”

The transduction and cell culture techniques used to generate and expand PD-L1 CAR haNKs are proprietary. The following text has been added to the Materials and methods section: “haNK cells (Jochems et al., 2016) transduced with a second-generation CAR targeting PD-L1 produced through proprietary transduction and cell culture techniques were obtained from NantKWest through a Cooperative Research and Development Agreement.”

2) In the first in vivo experiment, in particular the authors show that in the WT mice bearing MOC1 tumors, the CD8 depletion abrogated completely tumor growth inhibition. They attribute this effect to CD8 T cells being the primary source of INFγ driving the tumor cell PD-L1 expression in the model. This observation somehow conflicts with results in Figure 2 showing 50% killing of MOC1 cells without INF. In my opinion it could also be that the PD-L1 CAR haNK cells are binding PD-L1 but not killing the cells yet the blocking of PD-L1 is allowing CD8 T cells to kill the tumor cells, explaining why when CD8 T cells were depleted tumor inhibition was not seen anymore. To claim that the reason tumor inhibition was abrogated after CD8 depletion was due to INFγ, the authors should re-do the experiment with a group of mice that are CD8 depleted and treated with PD-L1 CAR haNK and intratumoral INFγ. This would then show that the PD-L1 CAR haNK cells are able to kill in vivo and not acting similarly to a blocking antibody.

Further investigation into the underlying mechanism behind the interaction between CD8 positive cells and the PD-L1 CAR haNKs is a great idea and we thank the reviewer for this suggestion. Although the reviewer poses an interesting experiment – we have tried local type II IFN injection into tumors in past unrelated experiments and didn’t have good results. We performed an alternative experiment, which we feel answers the same fundamental question raised by this astute reviewer. To address this critique, WT mice bearing MOC1 tumors were treated PD-L1 CAR haNKs with or without IFNγ depletion or TNFa depletion alongside CD8 depletion. IFNγ depletion abrogated the anti-tumor effect of PD-L1 CAR haNKs to a similar degree as CD8 cell depletion, whereas TNFa depletion did not. These data suggest that IFNγ is the link between CD8 cell depletion and abrogation of the effect of PD-L1 CAR haNKs. This experiment is now included in the new Figure 4C, with appropriate changes to the Materials and methods section.

The following text has been added to the Results section: “To determine if IFNγ is the major driver of PD-L1 being targeted by PD-L1 CAR haNKs and if CD8 T cells are the major source of IFNγ, treatments were repeated in the presence or absence of antibody depletions. […] These data suggested that IFNγ produced by CD8 T cells was the major driver of tumor cell PD-L1 expression being targeted by PD-L1 CAR haNKs.”

3) Various autoimmune symptoms are a major side effect of immune checkpoint inhibitors in the clinic. The availability of an immunocompetent mouse model may allow the interrogation of this phenomenon. Did these mice display signs of autoimmunity? In this paper there has not been experiments commenting on the toxicity of the therapy and this somehow gives a truncated story. Some toxicity studies should be added to complete the story or alternatively, the authors may present the rationale why such symptoms are not expected to arise in this experimental setting, perhaps due to the short time span of the experiment.

The inclusion of select blood chemistries and mouse body weights as measures of possible toxicity are a great suggestion and we apologize for not including these data previously. To address this critique, we have now included mouse body weight for mice treated with intravenous or intraperitoneal PD-L1 CAR haNKs and select blood chemistries for mice treated intraperitoneally. These data demonstrating no differences between control and PD-L1 CAR haNK treated mice. These data are included in the new Figure 4—figure supplement 1. The Materials and methods section has been changed accordingly.

The following text has been added to the Results section: “No weight loss, alterations in select blood chemistries, or other measurable forms of toxicity were observed with systemic PD-L1 CAR haNK treatment administered via intravenous or intraperitoneal routes (Figure 4—figure supplement 1).”

4) The kinetics of PD-L1 haNK cells in the immunocompetent mouse system are important to describe. The authors demonstrate that PD-L1 CAR haNK cells infiltrate MOC1 tumors at 1 hour after cell infusion but that they disappear by 24 hours. What is their kinetics in the blood, and in the spleen?

This is a reasonable request and we agree this information would add to the manuscript. We already had these data available and apologize for not including them in the original manuscript. We were able to quantify human NK cells in the peripheral blood and spleens of WT B6 mice bearing MOC1 tumors by flow cytometry. To address this critique, flow cytometric quantification of human NK cells in the blood and spleens of these mice at 1h and 24h is now included in the new Figure 5—figure supplement 1. This new data indicates that accumulation of PD-L1 CAR haNKs in the blood and spleen of mice is significantly decreased by 24 h compared to 1 h after adoptive transfer, similar to the results observed in tumor. We have also added the following text to the Results section: “Flow cytometric analysis 1 hour after treatment verified the presence of adoptively transferred PD-L1 CAR haNKs in the blood, spleens and tumors of MOC1 tumor bearing mice (Figure 5—figure supplement 1).”

5) Strong limitation of the murine models is represented by the fact that the authors have performed a subcutaneous flank injection of tumor cells, and not have provided activity in a orthotopic tumor model for HNSCC. This is a special requirement for HNSCC since the relevance of lymph node metastasis in this tumor model. An orthotopic tumor model could answer if haNKs CAR are able to migrate to the metastic and primary tumor site. Moreover, since PDL1 is expressed in several healthy tissues, a systemic iv infusion of the haNKs CAR-PDL1 need to be performed, to verify whether on-target off-tumor toxicity could limit the clinical application of the approach.

We agree that treatment of mice bearing an orthotopic cancer with metastatic potential with PD-L1 CAR haNKs would be very interesting and potentially provide insight into the ability of these cells to limit metastatic development or progression. However, the syngeneic model chosen for this project (MOC1) and the xenograft model chosen for this project (UMSCC-1) do not metastasize when implanted orthotopically or in the flank. We would argue that the study of therapeutic efficacy in metastases is beyond the scope of this specific project which is a “proof of concept” and first publication with PD-L1 CAR haNKs, but agree that it is a limitation of our work presented here. To address this critique and provide acknowledgement that this is a strong limitation of our study, we have added the following text to the Discussion section: “A major limitation of the conclusions of our data was the use of subcutaneous flank xenograft and syngeneic models that do not metastasize. The use of an orthotopic metastatic model transplanted into the oral cavity region could provide valuable insight into the ability of PD-L1 CAR haNKs to prevent or delay the development or the progression of metastases. This requires further study.”

With regard to IV vs. IP toxicity, we agree this is important data to include in the manuscript. To address this critique, we have now included in new Figure 4—figure supplement 1 mouse weights with either systemic IV or systemic IP PD-L1 CAR-haNK treatment of MOC1 tumor-bearing WT B6 mice. Neither IP nor IV treatment resulted in weight loss compared to control mice. This resulted in revision of the text as detailed above in the response to #3.

6) Impedance analysis is suitable only for short term ability of haNKs CAR-PDL1 to control the tumor. Long-term culture of at least 72hrs is required to verify antitumor control, even considering that haNKs have been irradiated.

We apologize for not including these data and agree with the reviewer that this is important data to include in this manuscript. Each impedance analysis experiment is always carried out to 120 hours (5 days), but typically the control cells start to lose impedance after 72 hours (likely due to nutrient depletion in the media in the impedance plate). Thus, to address this critique, each impedance plot displayed in Figure 1 and Figure 2 has been edited to now show impedance data out to 72 hours as suggested. These data show PD-L1 CAR haNK killing of IFN pre-treated target cell lines to be durable with no growth rebound. Additionally, the following text has been added to the Results section: “Killing of IFNγ pre-treated human target cells by PD-L1 CAR haNKs was durable as no tumor cell growth rebound was observed” and: “PD-L1 CAR haNK killing of IFN pre-treated murine target cells was less durable compared to human target cells, with a degree of rebound of target cell growth by 72 hours.”

7) In either Introduction or Discussion, authors have not been discussed recent and relevant paper in NK CAR field (e.g. Liu et al., 2018 and 2020; Ingegnere et al., 2019).

We agree that these references are useful to discuss. To address this concern, we have added the following text and references to the final paragraph of the Discussion section: “Based upon our data along with the data of others, CAR targeting of NK cells appears to enhance their killing efficacy (Ingegnere et al., 2019; Liu et al., 2018). To date, most progress has been made in the treatment of hematologic malignancy, with cord blood NK cells engineered to express a CD19-targeting CAR demonstrating high response rates in patients with relapsed CD19^+^ leukemia or lymphoma (Liu et al., 2020).”

8) Figure 3—figure supplement 1 shows the expression pattern of PD-L1 knockout human UMSCC-1, human UMSCC-11A and murine MOC1 cells. The MFI is significantly reduced in the PD-L1 knockout cells, however the expression seems not completely negative. The authors should show Isotype controls to clarify whether is only a matter of setting of the flow-analysis or if after the gene editing, the PDL1 has a dim expression that CAR.PDL1 is unable to recognize.

Thank you for pointing out this oversight – we agree isotype staining is important to add for interpretation of results. To address this critique, we have altered the bar graphs in new Figure 3—figure supplement 1 two now include the MFI for isotype controls for both PD-L1 and MHC class I staining. We now had to split the y-axis into two segments to allow the reader to see the actual bars for the PD-L1 ko and isotype control conditions. There was no statistically significant difference between the PD-L1 ko cells treated with IFN and the isotype control staining for each of the three cell lines.

The following text was added to the Results section: “Exposure of these PD-L1 knockout cells to IFNγ demonstrated a lack of baseline and IFNγ-inducible PD-L1 expression (compared to isotype control staining) but retained baseline and IFNγ-inducible HLA class I expression, demonstrating that overall IFNγ responses remained intact (Figure 3—figure supplement 1).”

9) Figure 6—figure supplement 1 shows that PDL1 is upregulated only marginally in the tumor section. Maybe this is reflecting haNKs CAR-PDL1 localization. It would be interesting if the Authors could integrate the staining of NK cells in combination with PDL1.

We agree that this is likely the case in these NSG tumor bearing mice that have little to no endogenous immunity. However, we have been unable technically to histologically stain for PD-L1 CAR haNKs in tissues harvested form mice. We have tried in both xenograft and syngeneic models. Best evidence at this time indicates that fixation in preparation for immunofluorescence somehow prevents detection of the cells (since we are able to detect them by flow). This issue is not unique to this project or to our program, as we and others have been unable to develop and validate assays to stain (immunofluorescence) for NK cells in human HNSCC tissues in general.

To address this critique, the following text has been added to the Discussion section: “Another limitation of our study is our technical inability to detect PD-L1 CAR haNKs in resected malignant tissues by immunofluorescence. […] Work to develop validated assays to determine tumor special localization of NK cell products is ongoing.”